American Society for Microbiology | Microbiology Spectrum

# Prophage-mediated lysogenic conversion drives virulence evolution and genomic plasticity in *Streptococcus suis* serotype 9

Zhenglong Wen,[1] Yaqi Guo,[1] Yan Liang,[1] Yanfang Li,[1] Kexun Lian,[1] Rui Tang,[1] Shoupan Li,[1] Pei Zheng,[2] Yonggang Qu[1]

**ABSTRACT** *Streptococcus suis* serotype 9 is an emerging zoonotic pathogen threatening pig production and public health. Here, we combined comparative genomics of 16 strains, including clinical isolate SS2401, to elucidate the role of prophages in shaping genomic plasticity and virulence evolution. The pan-genome was open (α = 0.375), highlighting extensive genetic diversity. Prophages were prevalent (56.25% of strains), significantly correlated with larger genomes, and exhibited two integration modes: direct insertion as genomic islands and integration at recombination hotspots associated with large-scale inversions. Phylogenetic analysis of the terminase large subunit (TerL) and whole-genome sequences revealed multiple independent acquisitions, with three prophages in SS2401 originating from distinct lineages ("one strain, multiple sources"). Recombination analysis detected 1,432 events across the core genome, indicating frequent horizontal gene exchange. The virulence gene *sly* (suilysin) was carried as a gene cassette within prophage Phi2401a. Notably, we identified integrase-deficient but otherwise intact prophages that may function as "gene prisons," stably fixing virulence traits. These findings demonstrate that prophages act as dual drivers of genomic architecture and as dynamic reservoirs for virulence genes, providing a framework for understanding bacterial adaptation and informing surveillance strategies in the swine industry.

**IMPORTANCE** *Streptococcus suis* serotype 9 poses a significant threat to pig farming and public health worldwide. This study reveals that prophages are not passive passengers but active architects of genomic plasticity and virulence evolution in this pathogen. We demonstrate that prophages contribute to genome expansion, facilitate large-scale chromosomal rearrangements, and carry key virulence genes such as *sly*. The discovery of a "one strain, multiple sources" acquisition pattern and the "gene prison" model explains how virulence traits can be stably maintained and disseminated. These findings redefine prophages as dual drivers of bacterial evolution, providing new insights for surveillance and intervention strategies in the swine industry.

**KEYWORDS** prophage, lysogenic conversion, *Streptococcus suis* serotype 9, virulence evolution, genomic plasticity

$S$treptococcus suis is a significant zoonotic pathogen that causes substantial economic losses in the global swine industry and poses a notable public health threat to occupationally exposed individuals (1–3). Currently, based on differences in capsular polysaccharide antigens, *S. suis* can be classified into 29 serotypes, which exhibit marked variations in geographical distribution, prevalence, and pathogenic potential (4). While serotype 2 is the most extensively studied, other serotypes such as 1, 7, 9, 14, and 1/2 are also recognized as highly pathogenic and are frequently associated with disease outbreaks in pigs and human infections (5–7).

**Peer Reviewer** Guoqiang Zhu, Yangzhou University, Yangzhou, China

Address correspondence to Pei Zheng, 18963862980@sohu.com, or Yonggang Qu, quyonggang@shzu.edu.cn.

Zhenglong Wen and Yaqi Guo contributed equally to this article. Author order was determined by mutual agreement.

The authors declare no conflict of interest.

Epidemiological data indicate that *S. suis* serotype 9 is among the predominant pathogenic serotypes in pig populations across Europe, North America, and Asia (8–10). Compared with classical, highly virulent strains such as serotype 2, serotype 9 strains exhibit greater strain-to-strain heterogeneity in virulence phenotypes (11). This diversity strongly suggests significant genetic differentiation and adaptive evolution within the *S. suis* serotype 9 population. A core driver of such evolution is bacterial genomic plasticity—the capacity of a genome to rapidly generate variation through the acquisition, loss, or rearrangement of genetic material (12). A key manifestation of this plasticity is an open pangenome structure, comprising a small set of core genes and a larger pool of accessory genes, which serves as the genetic foundation for adaptation and acquisition of new traits (13). Horizontal gene transfer (HGT), mediated by mobile genetic elements (MGEs), is a central force driving the continuous renewal of this accessory gene pool (14).

Among various MGEs, prophages—the integrated genomes of temperate bacteriophages—play a key role in bacterial evolution by mediating lysogenic conversion (15). Recent studies have shown that prophages can carry virulence factors (e.g., toxins, adhesins), antibiotic resistance genes, and fitness determinants, thereby enhancing host adaptability (16, 17). In *Staphylococcus aureus*, prophages are well-known contributors to pathogenicity (18). However, in S. *suis*, systematic investigations of prophage distribution, diversity, and functional impact have only recently emerged. Genomic surveys have revealed widespread prophage elements in *S. suis* populations, with some carrying virulence-associated genes such as suilysin (*sly*) and muramidase-released protein (*mrp*) (19–21). Nevertheless, the specific role of prophages in driving genomic plasticity and virulence evolution of *S. suis* serotype 9 remains poorly understood, highlighting the need for a comprehensive comparative genomic analysis.

In this study, we sequenced the complete genome of a clinical *S. suis* serotype 9 isolate (SS2401) and compared it with 15 publicly available genomes of the same serotype. Pangenome analysis revealed the genetic structure and virulence factor distribution within this population. We systematically identified prophages, assessed their association with genome size, and examined their impact on genomic plasticity via whole-genome alignment. Phylogenetic analysis of the terminase large subunit (TerL) traced prophage origins, while functional annotation uncovered prophage-encoded virulence genes and modular diversity. By integrating these approaches, this study elucidates the role of prophages as key drivers of genomic plasticity and virulence evolution in *S. suis* serotype 9.

## MATERIALS AND METHODS

### Bacterial strains and genomic data

A total of 16 *S. suis* serotype 9 strains were selected for this study to resolve their population genetics and prophage distribution (Table 1). The clinical isolate SS2401 was obtained in 2024 from the lung tissue of a diseased piglet from a farm in Xinjiang, China. It was confirmed as *S. suis* serotype 9 by 16S rRNA gene sequencing (22) and serotype-specific multiplex PCR (23). The 15 publicly available genomes were selected to represent the global diversity of *S. suis* serotype 9, encompassing strains from Europe (Spain, the Netherlands), North America (United States, Canada), and Asia (China), with isolation dates ranging from 2014 to 2020 and covering multiple sequence types (STs) including ST16, ST123, ST220, and others (Table 1). This selection aimed to capture both widespread clonal lineages and regionally restricted genotypes, providing a comprehensive epidemiological context for comparative genomics (8, 24–27).

### Genome sequencing and assembly

Strain SS2401 was revived on Tryptic Soy Agar (TSA; Haibo Biotech, Qingdao, China) supplemented with 5% fetal bovine serum (FBS; ExCell Bio, Suzhou, China). A single

**TABLE 1** Information on *S. suis serotype 9* strains used in this study

| Strains | Genomic accession | Serotype | ST | Country | Date | Host | Reference |
|---|---|---|---|---|---|---|---|
| 9401240 | LR738724.1 | 9 | 220 | Dutch | 2020 | Diseased pig | (26) |
| GD-0088 | LR738724.1 | 9 | 16 | Dutch | 2020 | Diseased pig | (26) |
| SS2401 | CM127139.1 | 9 | 16 | China | 2024 | Diseased pig | This study |
| Ss_109 | CP139876.1 | 9 | 16 | Spain | 2018 | Diseased pig | (8) |
| Ss_84 | GCA_034317195.1 | 9 | 123 | Spain | 2019 | Diseased pig | (8) |
| Ss_51 | GCA_034297355.1 | 9 | 1626 | Spain | 2014 | Diseased pig | (8) |
| Ss_106 | GCF_034317075.1 | 9 | 123 | Spain | 2018 | Diseased pig | (8) |
| DN13 | CP015557.1 | 9 | 234 | Canada | 2016 | pig | (24) |
| D12 | CP002644.1 | 9 | 619 | Canada | 2014 | pig | (24) |
| 2022WUSS001 | GCF_028863525.1 | 9 | 1198 | China | 2021 | Healthy pig | (27) |
| 2022WUSS046 | GCF_028863345.1 | 9 | 220 | China | 2021 | Healthy pig | (27) |
| 40455 | JAASCV000000000 | 9 | 790 | United States | 2016 | pig | (25) |
| 40462 | JAASDC000000000 | 9 | 1198 | United States | 2017 | pig | (25) |
| 40468 | JAASDI000000000 | 9 | 1206 | United States | 2017 | pig | (22) |
| 40521 | JAASDO000000000 | 9 | 790 | United States | 2017 | pig | (22) |
| 40529 | JAASDV000000000 | 9 | 1210 | United States | 2016 | pig | (22) |

colony was inoculated into Tryptic Soy Broth (TSB; Haibo Biotech) with 5% FBS and incubated overnight at 37°C with shaking (180 rpm). The culture was then diluted 1:100 into fresh TSB with 5% FBS and grown to mid-logarithmic phase. Cells were harvested by centrifugation (4°C, ≥3 g wet weight) and submitted to Novogene Co., Ltd. (Beijing, China) for DNA extraction and sequencing. Genomic DNA was extracted using the STE method, assessed by agarose gel electrophoresis, and quantified with a Qubit fluorometer. Whole-genome sequencing was performed on a PacBio Sequel IIe platform (SMRT sequencing). Low-quality reads were filtered using SMRT Link v8.0 and assembled *de novo* with Canu v2.0, yielding a single gapless contig.

## Pangenome and core genome analysis

The multilocus sequence types (MLSTs) of the 16 *S. suis* serotype 9 strains were determined using the PubMLST database (28). Genomes were annotated with Prokka v1.14.6 (29). Pangenome analysis was performed using Roary v3.13.0 with a 99% protein identity threshold; core genes were defined as present in ≥95% of strains (30). A maximum-likelihood phylogenetic tree was constructed from the core gene alignment in Geneious Prime using 1,000 bootstrap replicates (31). The tree was visualized with iTOL v5 (32), along with a binary matrix of 22 virulence factors identified by BLASTP against VFDB (33). To quantitatively assess pan-genome openness, accumulation curves were generated by randomly permuting the gene presence/absence matrix 100 times for both 99% and 95% identity thresholds. Heaps' law parameters ($\alpha$) were calculated using custom R scripts (34, 35). For functional classification, representative protein sequences from all gene families were annotated with eggNOG mapper v2 against the COG database (36). To determine whether phylogenetic incongruence could be attributed to recombination, we applied Gubbins to the core genome alignment (37).

## Prophage identification and genomic impact

Prophage regions were predicted using PHASTER (38, 39). Strains harboring at least one intact prophage (score >90) were considered prophage-positive. The relationship between prophage carriage and genome size was assessed using the Mann-Whitney *U* test (GraphPad Prism v10.2.3). Sensitivity analysis excluded two unusually small genomes (Ss_106 and Ss_51). Continuous variables (prophage total length, count, PHASTER score) were correlated with genome size using Spearman's rank correlation and linear regression (20).

To investigate genomic rearrangements, whole-genome alignments of three ST16 strains (SS2401, GD-0088, Ss_109) were performed using Mauve v2.0 (40). Locally collinear blocks and crossing patterns were examined to infer prophage-associated structural variants.

## Phylogenetic analysis of prophages

Amino acid sequences of the terminase large subunit (TerL) were extracted from the 11 intact prophages. Reference TerL sequences from other *S. suis* serotypes and an outgroup (Bacillus phage JBP901, NC_027352.1) were obtained from GenBank. Sequences were aligned with MAFFT v7.520 (L-INS-i), and a maximum-likelihood tree was constructed in MEGA12 using the LG+G model with 1,000 bootstrap replicates (41). The tree was rooted with the outgroup and visualized in iTOL v6. To complement the TerL analysis, a whole-genome nucleotide tree of the 11 prophages was built from MAFFT alignments using IQ-TREE 2.2.0 with the best-fit model (GTR+F + I+G4) and 1,000 ultrafast bootstrap replicates (42).

## Functional annotation and comparative genomics of prophage-associated genes

Prophage gene products were searched against VFDB and PHI-base using BLASTP (E-value ≤ 1e-5) (33, 43). Hits with ≥30% identity and ≥50% query coverage were considered candidates; for key virulence genes (e.g., sly), a higher threshold of ≥80% identity was applied. Synteny alignment and visualization of the 11 intact prophages were performed with Clinker v0.0.27 (protein identity threshold 30%) (44). To independently validate the predicted boundaries of the sly-carrying prophage Phi2401a, we searched for phage attachment (att) sites in the regions immediately flanking the prophage.

## RESULTS

### Pangenome structure and core genome-based phylogenetic analysis

Pangenome analysis of the 16 strains identified 5,710 gene clusters, with 703 core genes (12.3%, present in ≥95% of strains) and 5,007 accessory genes (87.7%) (Table 2). Accumulation curves and Heaps' law fitting confirmed an open pan-genome, with α = 0.375 (99% identity, 95% CI: 0.372–0.379). Sensitivity analysis using a 95% threshold gave a similar result (α = 0.327, 95% CI: 0.324–0.330; Fig. S1).

The core-genome phylogenetic tree grouped strains primarily by ST (Fig. 1). ST16 (SS2401, GD-0088, Ss_109), ST123 (Ss_84, Ss_106), and ST790 (40455, 40521) each formed well-supported clades. However, two ST220 strains (9401240 and 2022WUSS046) were placed distantly from each other (Fig. 1). To determine whether this incongruence could be attributed to recombination, we performed recombination detection using Gubbins on the core-genome alignment. The analysis identified a total of 1,432 recombination events spanning 490,832 bp (average fragment length 343 bp). After masking recombinant regions, the reconstructed tree (Fig. S2) showed that the two ST220 strains clustered together, in contrast to their distant placement in the core genome tree. This suggests that recombination contributed to their initial phylogenetic separation. However, other topological adjustments were minor, indicating that

**TABLE 2** Pangenome statistics for *S. suis* serotype 9

| Category | Definition (% of strains) | Number of genes | Percentage of total |
|---|---|---|---|
| Core genome | 95%–100% | 703 | 12.3% |
| Shell genome | 15%–94% | 2,199 | 38.5% |
| Cloud genome | 0%–15% | 2,808 | 49.2% |
| Accessory genome | Shell + Cloud | 5,007 | 87.7% |
| Pan-genome | 0%–100% | 5,710 | 100% |

A

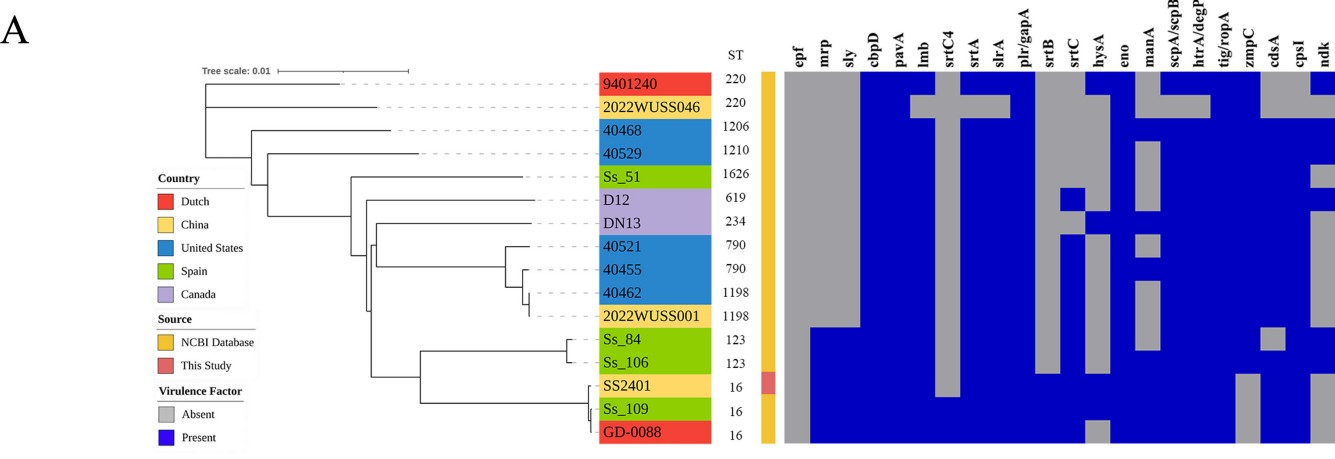

B

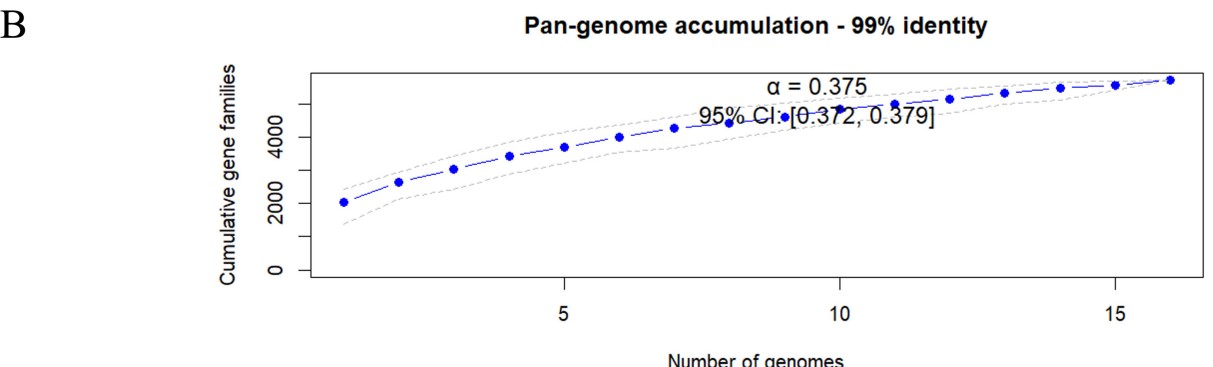

FIG 1 Genomic diversity and pan-genome openness of *Streptococcus suis* serotype 9. (A) Integrated analysis of phylogeny, STs, and virulence gene profiles of 16 *S. suis* serotype 9 strains. (B) Pan-genome accumulation curve based on a 99% amino acid identity threshold.

recombination plays a role in genetic diversification but does not account for all observed phylogenetic inconsistencies.

Analysis of geographical origins revealed evidence of intercontinental spread. The ST16 clade, for instance, included strains from China, the Netherlands, and Spain and exhibited very short branch lengths, suggesting it may represent a recently disseminated, successful clone. Multiple strains from the United States (e.g., 40455, 40462, 40468) were scattered across different phylogenetic branches, indicating that the local population likely originated from multiple independent introduction events. Furthermore, the ST220 lineage clustered at the base of the tree with unique novel lineages such as ST1206 and ST1210, which may represent ancient or regionally restricted ancestral genetic backgrounds within the population.

### Genome functional landscape and virulence gene distribution

COG functional analysis revealed distinct roles for core and accessory genes (Table S1). Core genes were enriched for essential functions, such as translation (J, 18.9%), transcription (K, 11.7%), replication and repair (L, 10.5%), and cell wall biogenesis (M, 9.1%). In contrast, accessory genes showed higher proportions in adaptation-related categories, including defense mechanisms (V, 2.7% vs. 4.4% in core), signal transduction (T, 1.6% vs. 3.6%), carbohydrate metabolism (G, 3.6% vs. 14.9%), and inorganic ion transport (P, 1.6% vs. 9.4%). Notably, 16.3% of accessory genes lacked COG assignments, suggesting potential niche-specific functions.

Beyond these general functional patterns, we examined the distribution of 22 virulence-associated genes, which exhibited a dual structure comprising a "core

virulence gene set" and a "variable accessory gene set" (Fig. 1A). The core set (e.g., *cbpD*, *pavA*, *eno*, *ropA*) was present in ≥93.8% of strains, indicating a fundamental role in basic fitness, likely maintained through vertical inheritance. The variable set (e.g., *cpsI*, *manA*, *srtB*) exhibited a mosaic distribution not strictly correlated with ST, suggesting frequent horizontal transfer or lineage-specific loss. Notably, the *mrp* and *sly* genes were strictly co-localized and conserved in all ST16 and ST123 strains, but absent in other STs. Although ST16 and ST123 belong to distinct phylogenetic branches (Fig. 1A), their shared virulence module suggests either independent acquisition via separate horizontal transfer events or inheritance from an ancient common ancestor followed by lineage divergence.

## Prophage distribution and its impact on genome size

Whole-genome prophage prediction using PHASTER identified 11 intact prophages across 9 of the 16 strains (56.25%), with the clinical isolate SS2401 harboring three intact prophages, the highest among all strains (Table 3 and Fig. 2A). To assess the relationship between prophage carriage and genome size, we compared genome sizes between strains with and without intact prophages. Prophage-positive strains had significantly larger genomes (median 2.206 Mb, IQR: 2.175–2.286 Mb; $n = 9$) than prophage-negative strains (2.065 Mb, IQR: 1.794–2.142 Mb; $n = 7$) (Mann-Whitney $U = 4$, $P = 0.011$) (Fig. 2B).

Two strains (Ss_106 and Ss_51) had exceptionally small genomes (<1.8 Mb) and unusually low GC content, raising the possibility that they might bias the comparison. A sensitivity analysis excluding these potential outliers confirmed that the significant difference persisted (positive group median: 2.206 Mb, $n = 9$; negative group median: 2.073 Mb, IQR: 2.049–2.169 Mb; $n = 5$; Mann-Whitney $U = 6$, $P = 0.014$), indicating that the result is not driven by these outliers.

We further examined prophage load as a continuous variable. The total length of intact prophage sequences per strain showed a significant positive correlation with genome size (Spearman's $\rho = 0.619$, $P = 0.0105$). Similar correlations were observed for prophage count ($\rho = 0.644$, $P = 0.0071$) and total PHASTER score ($\rho = 0.640$, $P = 0.0075$). These findings demonstrate that prophage load is significantly associated with genome size in a dose-dependent manner, and this association is robust to alternative definitions and outlier exclusion.

**TABLE 3**  Genomic features and prophage content of *S. suis* serotype 9 strains

| Strains | Size (bp) | GC content (%) | No. of total CDSs | Intact prophages | Incomplete prophages |
|---|---|---|---|---|---|
| 9401240 | 2,195,215 | 41.43 | 2,036 | 0 | 0 |
| GD-0088 | 2,298,012 | 41.2 | 2,213 | 1 | 0 |
| SS2401 | 2,449,539 | 41.00 | 2,488 | 3 | 0 |
| Ss_109 | 2,274,578 | 41.00 | 2,275 | 1 | 0 |
| Ss_84 | 2,240,275 | 41.00 | 2,206 | 1 | 0 |
| Ss_51 | 1,794,393 | 40.00 | 1,762 | 0 | 0 |
| Ss_106 | 1,462,712 | 38.00 | 1,415 | 0 | 0 |
| DN13 | 2,142,184 | 41.50 | 2,102 | 0 | 0 |
| D12 | 2,183,059 | 41.50 | 2,183 | 1 | 1 |
| 2022WUSS001 | 2,064,590 | 39.00 | 2,037 | 0 | 0 |
| 2022WUSS046 | 2,203,815 | 42.00 | 2,176 | 1 | 1 |
| 40455 | 2,033,844 | 41.38 | 1,939 | 0 | 0 |
| 40462 | 2,073,317 | 41.16 | 1,984 | 0 | 0 |
| 40468 | 2,167,241 | 41.30 | 2,045 | 1 | 0 |
| 40521 | 2,092,795 | 41.32 | 2,032 | 1 | 1 |
| 40529 | 2,205,574 | 41.10 | 2,102 | 1 | 0 |

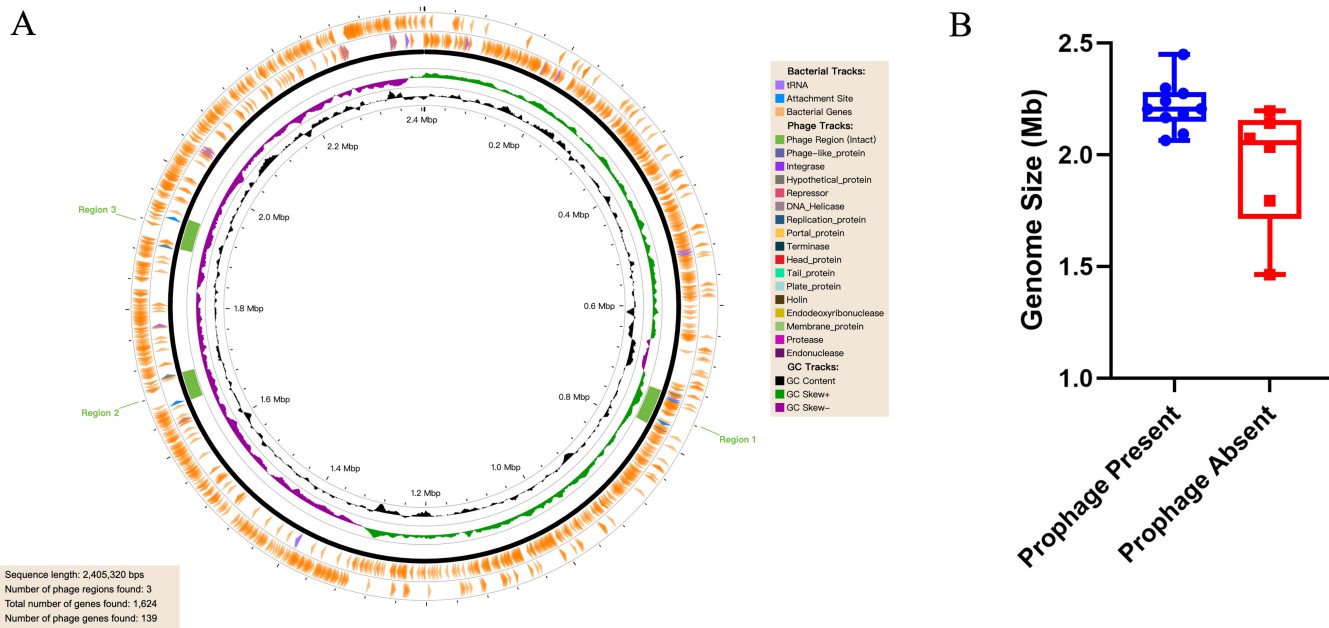

**FIG 2** Prophage integration sites in strain SS2401 and the impact of prophage on genome size. (A) Whole-genome circle map of strain SS2401. The positions of the three intact prophages (phi2401a, b, c) are highlighted in green. (B) Comparison of genome sizes between prophage-positive ($n = 9$) and prophage-negative ($n = 7$) *S. suis* serotype 9 strains.

## Genomic characteristics of intact prophages and their impact on host genome structure

A total of 11 intact prophages were identified among the 16 *S. suis* serotype 9 strains, and their basic genomic features are summarized in Table 4. The genome sizes of these prophages ranged from 31.1 Kb to 54.5 Kb, with an average GC content of approximately 41.1%. To investigate the macroscopic impact of these prophages on the host genome, whole-genome alignment was performed on three strains (SS2401, GD-0088, and Ss_109) belonging to the same clonal group (ST16) and all harboring intact prophages (Fig. 3). The analysis revealed two distinct modes of association between prophage integration and genomic structure. Mode 1: Direct insertion as a genomic island. In SS2401, the integration sites of phi2401a and phi2401b were located within locally collinear blocks (LCBs). Alignment of this strain with a strain lacking the corresponding prophage (e.g., GD-0088) showed that these integration sites corresponded to sequence deletions within the respective LCBs in the GD-0088 genome. This indicates that such prophages were directly inserted as complete genomic islands into continuous sequence regions of the host genome via site-specific recombination, without directly inducing large-scale rearrangements in the flanking sequences. Mode 2: Prophage integration associated with genomic rearrangement. In contrast to Mode 1, the integration site of prophage phiGD-0088 in strain GD-0088 was located at the boundary between two LCBs. Synteny analysis revealed that the order of the homologous blocks corresponding to these two LCBs in strains SS2401 and Ss_109 differed from that in GD-0088, with clear crossings of collinear lines (Fig. 3). This pattern is consistent with the occurrence of genomic inversion or translocation events in this region. The observed association suggests that the integration site of phiGD-0088 (or its flanking sequences) may have served as a recombination hotspot, potentially contributing to large-scale structural rearrangements in the surrounding genomic region.

**TABLE 4** Basic characteristics of the intact prophages identified in this study

| Prophage | Host strain | Completeness (score) | Genome size (Kb) | Region position (bp) | Predicted genes | GC, % |
|---|---|---|---|---|---|---|
| phiGD-0088 | GD-0088 | Intact (140) | 47.6 | 2,054,232–2,101,845 | 51 | 41.67 |
| phi2401a | SS2401 | Intact (150) | 54.5 | 731,183–785,686 | 65 | 40.72 |
| phi2401b | SS2401 | Intact (150) | 43.9 | 1,658,861–1,702,790 | 59 | 39.85 |
| phi2401c | SS2401 | Intact (100) | 47.1 | 1,891,745–1,938,875 | 58 | 41.49 |
| phiSs_109 | Ss_109 | Intact (110) | 38.1 | 1,459,052–1,497,159 | 51 | 41.94 |
| phiSs_84 | Ss_84 | Intact (110) | 40.4 | 622,521–662,965 | 62 | 40.54 |
| phiD12 | D12 | Intact (150) | 44.3 | 1,166,786–1,211,131 | 43 | 40.10 |
| phi40468 | 40468 | Intact (130) | 38.3 | 99,123–137,508 | 52 | 41.74 |
| phi40529 | 40529 | Intact (150) | 44.5 | 2,082,205–2,126,748 | 71 | 41.69 |
| phi40521 | 40521 | Intact (100) | 49.3 | 443,681–492,906 | 63 | 41.11 |
| phi2022WUSS046 | 2022WUSS046 | Intact (150) | 31.1 | 1,189,176–1,220,351 | 50 | 41.01 |

## Phylogenetic analysis of prophages based on the terminase large subunit

To trace the evolutionary origins of the prophages, a maximum likelihood phylogenetic tree was constructed based on TerL amino acid sequences, including the 11 intact prophages identified in this study, reference *S. suis* prophage sequences, and the Bacillus phage JBP901 as an outgroup (Fig. 4). The tree revealed that the three prophages from strain SS2401 (phi2401a, b, and c) did not form a monophyletic cluster but were distributed across three distinct, well-supported clades (bootstrap ≥0.62), providing strong evidence for a "one strain, multiple sources" acquisition pattern. Specifically, phi2401a clustered with phi2022WUSS046 and phiD12 (bootstrap 0.6262), Phi2401b grouped with phiST1 and phiNJ2 (bootstrap 0.6168), and Phi2401c formed a clade with phiSs_84, phi7917, and phi891591 (bootstrap 0.8411). The outgroup JBP901 was robustly positioned at the base of the tree, confirming the direction of evolution. The tree also

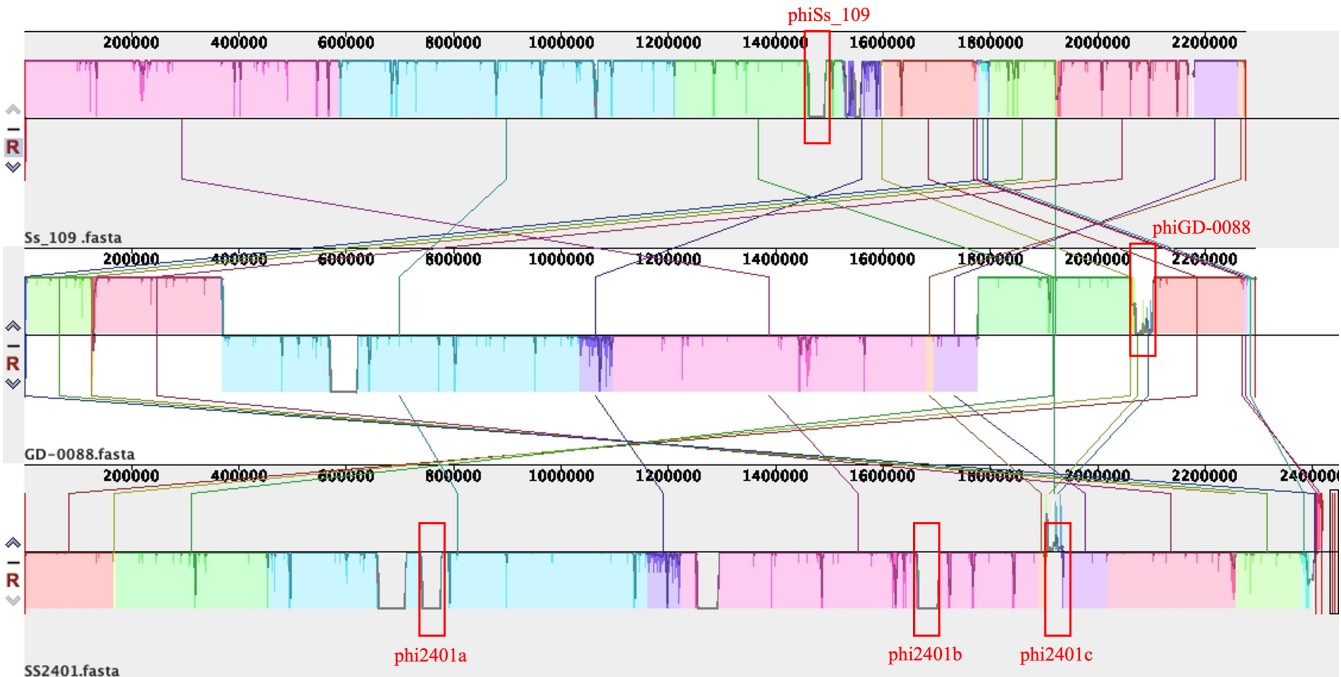

**FIG 3** Analysis of prophage integration modes based on whole-genome alignment. From top to bottom, the whole-genome alignments of strains Ss_109, GD-0088, and SS2401 are displayed. Red boxes mark the integration sites of intact prophages in each strain. Connecting lines represent homologous regions, with crossings of lines indicating genomic inversions.

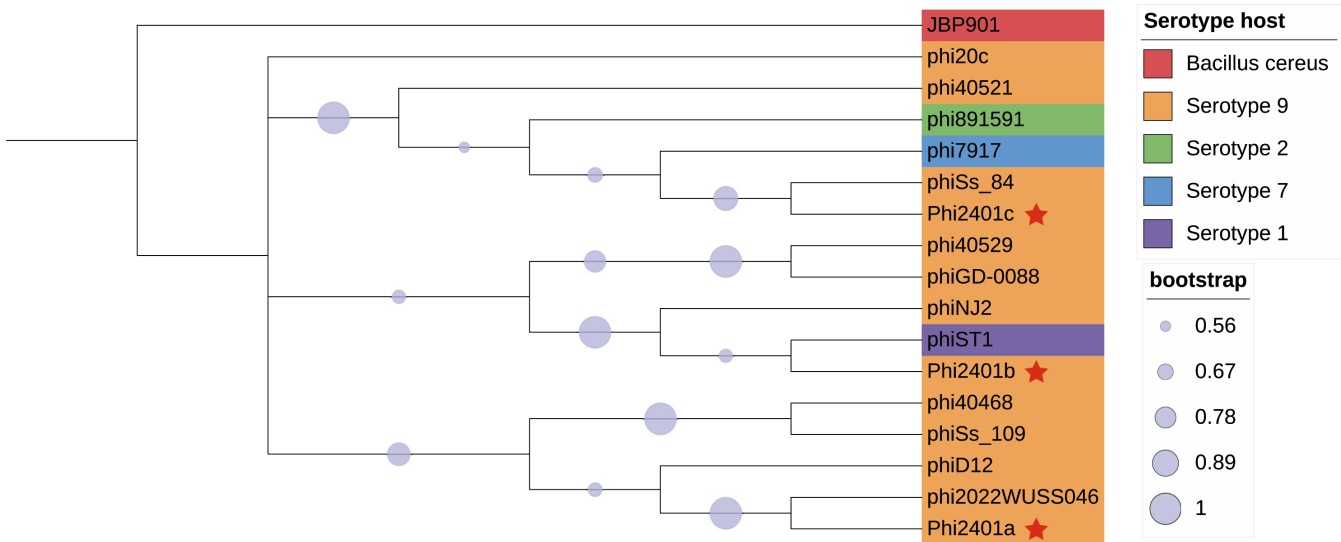

**FIG 4** Phylogenetic analysis of prophage TerL. Red stars mark the three prophages (phi2401a, b, c) originating from the same strain, SS2401.

showed that prophages from different *S. suis* serotypes were intermingled, indicating frequent horizontal transfer across serotypes.

To further substantiate the "one strain, multiple sources" hypothesis, we constructed a whole-genome nucleotide tree of the 11 intact prophages (Fig. S3). The topology was largely congruent with the TerL-based phylogeny: the three prophages from SS2401 (phi2401a, phi2401b, and phi2401c) again occupied distinct clades

## Functional annotation and comparative genomics of prophage-associated genes

Synteny alignment of the 11 intact prophages revealed substantial modular diversity (Fig. 5). Although all prophages possessed conserved core modules typical of phages (e.g., DNA packaging, head and tail assembly), the order and gene composition of these modules varied considerably. Notably, phiD12 and phi2022WUSS046 were classified as intact by PHASTER (score = 150) but lacked the integrase gene, while retaining all other essential functional modules (terminase, structural proteins, lysis components). This absence suggests they may be unable to excise, potentially becoming stably integrated "gene prisons" for associated genes.

Screening against VFDB and PHI-base identified known virulence or host-adaptation genes in a subset of prophages (Table 5). The *sly* gene in phi2401a exhibited 100% identity to the reference suilysin sequence, with full-length coverage, satisfying the high-confidence threshold (≥80%). Synteny analysis placed *sly* in a variable region downstream of the head gene module, flanked by sequences homologous to other prophages, consistent with a gene cassette insertion (Fig. 5). Consistent with its intact status, analysis of the phi2401a integration sites identified a 14-bp direct repeat (5′-T TTATGATATAATG-3′) precisely at the predicted boundaries (attL: 731,183–731,196; attR: 785,686–785,699). This canonical attachment site perfectly matches the PHASTER-predicted coordinates (731,183–785,686), providing definitive molecular evidence for the structural accuracy of phi2401a and confirming its annotation as an intact, functional prophage.

Further functional annotation analysis demonstrated that prophages can serve as essential platforms for acquiring functional genes, such as virulence factors, from other bacterial species or different strains of the same species. This was evidenced by the independent presence of the iron-uptake-related gene *feoB* (presumably derived from *Pseudomonas aeruginosa*) in two distinct prophages, phi2401a and phi40521; the adhesion gene *sspA* (originating from *S. suis* itself) being identified in the genetically

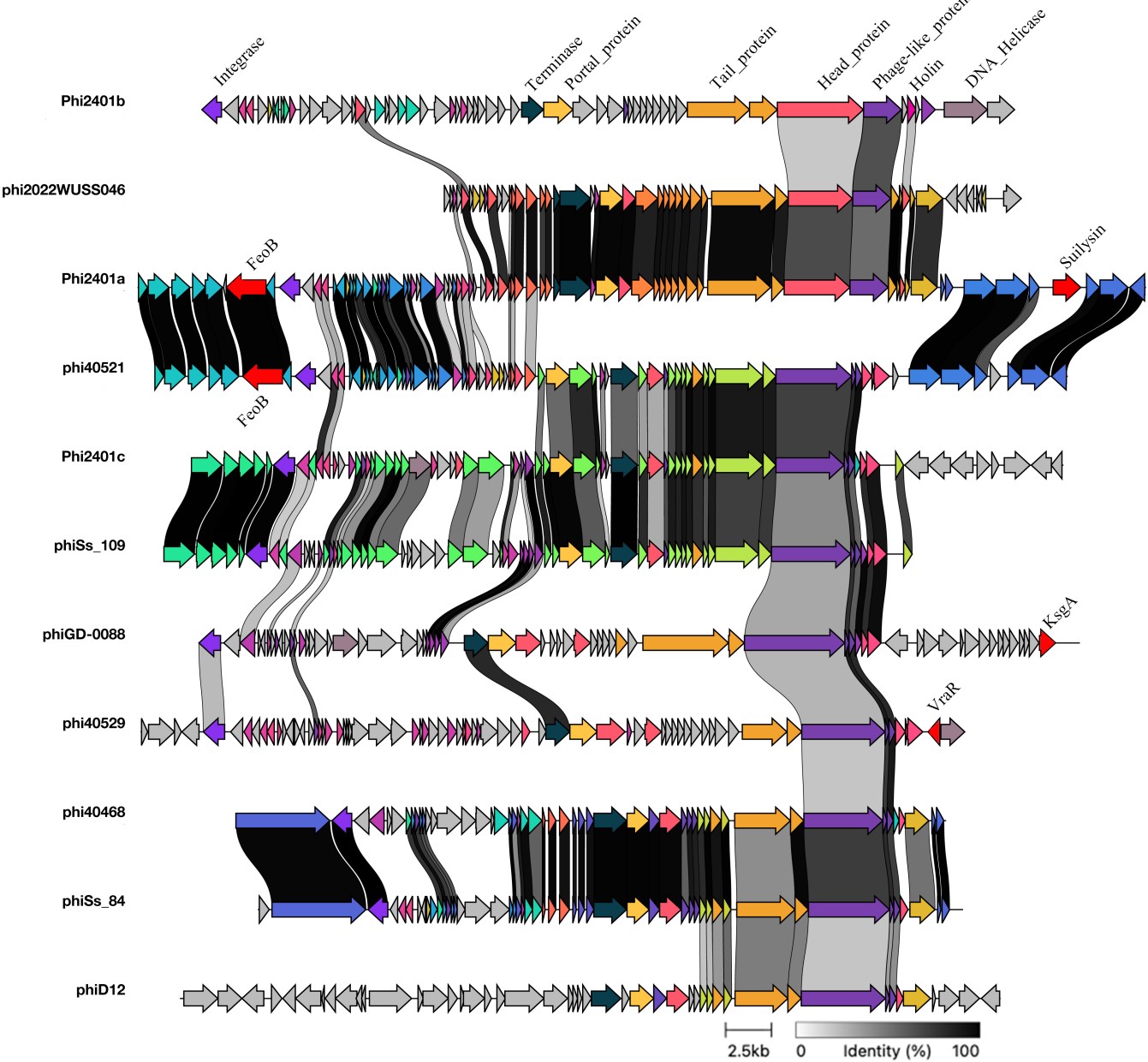

**FIG 5** Synteny alignment of the modular structures of the eleven intact prophage genomes. Arrows represent protein-coding sequences (CDSs). A red arrowhead indicates the *sly* gene (hemolysin) within Phi2401a. Prophages phiD12 and phi2022WUSS046 lack the integrase gene. The absence of integrase suggests these prophages may be permanently integrated, potentially acting as stable "gene prisons."

distant prophages phiSs_109 and phi40468; and prophage phiGD-0088 carrying the *ksgA* gene, which is associated with *Acinetobacter baumannii*. The detailed genomic structure of each intact prophage, including the presence and organization of key functional modules and virulence-associated genes, is summarized in Table S2.

## DISCUSSION

*Streptococcus suis* serotype 9 is an emerging zoonotic pathogen with a significant impact on the global swine industry (45, 46). Through integrated comparative genomics of 16 strains, this study systematically dissects the role of prophages in shaping genomic plasticity and virulence evolution of this serotype. Our findings demonstrate

**TABLE 5** Virulence and host adaptation genes identified within prophages

| Prophage | Virulence/adaptation gene | Homologous distribution | Pathogen species | Mutant phenotype | E-value | Identity (%) |
|---|---|---|---|---|---|---|
| phiGD-0088 | KsgA | Conserved in diverse bacteria (e.g., *Acinetobacter baumannii*, *Escherichia coli*) | *Acinetobacter baumannii* | Reduced virulence | 2.00E-44 | 96.00 |
| phi2401a | Sly | *Streptococcus suis* (species-specific) | *Streptococcus suis* | Reduced virulence | 0.00 | 100.00 |
| phi2401a | FeoB | Conserved in multiple bacterial species (e.g., *Pseudomonas aeruginosa*) | *Pseudomonas aeruginosa* | Loss of pathogenicity | 0.00 | 95.50 |
| phiSs_109 | SspA | *Streptococcus suis* (species-specific) | *Streptococcus suis* | Reduced virulence | 0.00 | 96.69 |
| phi40468 | sspA | *Streptococcus suis* (species-specific) | *Streptococcus suis* | Reduced virulence | 0.00 | 96.16 |
| phi40529 | VraR | *Streptococcus suis* (species-specific) | *Streptococcus suis* | Reduced virulence | 7.00E-67 | 100.00 |
| phi40521 | FeoB | Conserved in multiple bacterial species (e.g., *Pseudomonas aeruginosa*) | *Pseudomonas aeruginosa* | Reduced virulence | 1.00E-117 | 96.09 |

that prophages are not merely passive genetic cargo but active contributors to genome diversification, acting both as structural architects of chromosomal rearrangements and dynamic reservoirs of virulence-associated genes.

The open pan-genome architecture of serotype 9, quantitatively confirmed by Heaps' law ($\alpha = 0.375$), reflects extensive genetic diversity and provides a vast accessory gene pool for adaptation. The relatively low $\alpha$ value, however, indicates substantial genetic overlap among strains, likely due to the global dissemination of successful clonal lineages such as ST16 and ST123. This pattern is consistent with recent genomic surveys of *S. suis* serotype 9 in Spain and China, which also reported clonal expansion and lineage-specific gene content (8, 24). Similar observations have been made in other streptococcal pathogens; for example, a 2023 study on *S. agalactiae* revealed that prophage carriage contributes to lineage-specific accessory gene pools and niche adaptation (47).

Recombination analysis detected 1,432 events spanning nearly 0.5 Mb of the core genome, providing quantitative evidence for horizontal gene exchange in this population. The average recombinant fragment length (~343 bp) is consistent with recent estimates in *S. pyogenes* (300–400 bp) obtained using similar methods (48). Notably, after masking recombinant regions, the two ST220 strains clustered together (Fig. S2), indicating that recombination contributed to their initial phylogenetic separation. This finding underscores the importance of accounting for recombination when inferring evolutionary relationships, as recently emphasized in a 2023 comparative study of *S. suis* population structure (49).

Prophages were prevalent (56.25% of strains) and significantly associated with larger genome size in a dose-dependent manner, as evidenced by positive correlations between genome size and prophage total length (Spearman's $\rho = 0.619$, $P = 0.0105$), count ($\rho = 0.644$, $P = 0.0071$), and PHASTER score ($\rho = 0.640$, $P = 0.0075$). This relationship mirrors observations in *Escherichia coli*, where prophage load correlates with genome expansion (50, 51). Whole-genome alignment of three ST16 strains revealed two distinct integration modes: direct insertion as genomic islands (phi2401a/b) and integration at recombination hotspots associated with inversions (phiGD-0088). The latter suggests that prophage integration sites may serve as catalysts for large-scale rearrangements, a phenomenon recently documented in *Salmonella enterica* (52). These observations suggest that prophages may serve as catalysts for genomic plasticity across diverse bacterial pathogens.

Phylogenetic analysis of TerL and whole-genome sequences placed the three SS2401 prophages in distinct clades, supporting a "one strain, multiple sources" acquisition pattern. This mosaicism is increasingly recognized in other pathogens; for example,

a 2023 study of *Staphylococcus aureus* prophages revealed similar lineage mixing, highlighting the role of horizontal transfer in shaping prophage repertoires (53).

Functional annotation revealed that the *sly* gene is carried as a gene cassette within phi2401a, with 100% identity to the reference suilysin sequence. This finding directly demonstrates lysogenic conversion in *S. suis* serotype 9. It adds to a growing list of phage-encoded virulence factors documented in recent years. The presence of *feoB*, *sspA*, and *ksgA* in multiple prophages further illustrates the potential of these elements to acquire and disseminate diverse functional genes. The "gene prison" model proposed here, based on observations of integrase-deficient but otherwise intact prophages (phiD12 and phi2022WUSS046), offers a conceptual framework for understanding how virulence modules can become stably fixed within lineages. A similar phenomenon was recently described in *Staphylococcus aureus*, where a defective prophage was shown to have been co-opted as a stable genomic island carrying immune evasion genes (54). Whether such elements retain any residual function or serve solely as genetic relics warrants further experimental investigation.

The novelty of this work lies in the systematic integration of pan-genomics, recombination detection, and prophage-focused analyses, revealing the multifaceted roles of prophages in serotype 9 evolution. However, several limitations should be acknowledged. First, the findings are based on bioinformatic predictions; experimental validation (e.g., phage induction, gene knockout) is needed to confirm functional roles and inducibility. Second, the sample size, though geographically diverse, is limited to 16 strains; larger collections would enable finer phylogeographic inference. Third, the "gene prison" hypothesis requires direct testing through excision assays. Fourth, this study focused solely on serotype 9; whether similar dynamics occur in other serotypes remains to be explored.

Future research should combine long-read sequencing with experimental approaches to precisely map prophage boundaries and assess mobility. Functional studies of prophage-encoded genes, particularly *sly* and *feoB*, in relevant infection models (e.g., porcine macrophages, mouse models) would clarify their contribution to pathogenesis. Longitudinal surveillance of prophage content and virulence gene profiles across different regions and production systems could reveal ecological drivers of prophage dissemination. Addressing these questions will deepen our understanding of how prophages drive the adaptive evolution of *S. suis* and may inform surveillance and intervention strategies in the swine industry.

## Conclusion

This study demonstrates that prophages are key drivers of genomic plasticity and virulence evolution in *S. suis* serotype 9. The open pan-genome (α = 0.375) and frequent recombination (1,432 events) provide a dynamic genetic landscape, while prophage carriage correlates with genome expansion and contributes to structural variations through two distinct integration modes. Phylogenetic analyses revealed a "one strain, multiple sources" acquisition pattern. The discovery of integrase-deficient prophages suggests a "gene prison" model for stable virulence gene maintenance. COG functional classification further supports the accessory genome's role in adaptation. These findings establish prophages as dual architects of genome evolution and reservoirs for virulence traits, providing a framework for understanding bacterial adaptation and informing surveillance strategies in the swine industry. Experimental validation and broader sampling will be essential to confirm causal mechanisms and extend these insights to other serotypes.

## ACKNOWLEDGMENTS

This work was funded by the Natural Science Support Program Project of the Xinjiang Production and Construction Corps (2024DA007).

The authors are also grateful to Dr. Ahmad Ali and Mr. Uzair Alam for their valuable assistance in language editing and polishing of the manuscript.

## AUTHOR AFFILIATIONS

[1]College of Animal Science and Technology, Shihezi University, Shihezi, Xinjiang, China
[2]Xinjiang Tecon Animal Husbandry Technology Co. Ltd., Changji, Xinjiang, China

## AUTHOR ORCIDs

Pei Zheng  http://orcid.org/0009-0007-1491-8639
Yonggang Qu  http://orcid.org/0000-0003-2649-4144

## AUTHOR CONTRIBUTIONS

Zhenglong Wen, Data curation, Formal analysis, Investigation, Methodology, Software, Writing – original draft | Yaqi Guo, Investigation, Methodology, Visualization | Yan Liang, Conceptualization, Investigation, Resources, Validation | Yanfang Li, Conceptualization, Formal analysis, Investigation, Resources, Visualization | Kexun Lian, Formal analysis, Investigation, Validation | Rui Tang, Data curation, Investigation, Methodology | Shoupan Li, Data curation, Investigation, Methodology | Pei Zheng, Conceptualization, Formal analysis, Funding acquisition, Supervision, Validation | Yonggang Qu, Conceptualization, Formal analysis, Funding acquisition, Supervision, Validation, Writing – review and editing

## DATA AVAILABILITY

The whole-genome sequencing data of the newly isolated strain SS2401 generated in this study have been deposited in the NCBI GenBank database under accession number CM127139.1. The accession numbers for the genome sequences of the remaining 15 public strains used for comparative analysis are detailed in Table 1.

## ADDITIONAL FILES

The following material is available online.

### Supplemental Material

**Supplemental material (Spectrum00061-26-s0001.docx).** Fig. S1 to S3; Tables S1 and S2.

### Open Peer Review

**PEER REVIEW HISTORY (review-history.pdf).** An accounting of the reviewer comments and feedback.

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
