## [Reviewer comments · Microbiology Spectrum]

Microbiology Spectrum

Prophage-Mediated Lysogenic Conversion Drives Virulence Evolution and Genomic Plasticity in *Streptococcus suis* Serotype 9

Zhenglong Wen, Yaqi Guo, Yan Liang, Yanfang Li, Kexun Lian, Rui Tang, Shoupan Li, Pei Zheng, and Yonggang Qu

Corresponding Author(s): Yonggang Qu, Shihezi University

Review Timeline:

Submission Date:	January 7, 2026
Editorial Decision:	February 4, 2026
Revision Received:	March 13, 2026
Accepted:	March 16, 2026

Editor: Catherine Brissette

Reviewer(s): Disclosure of reviewer identity is with reference to reviewer comments included in decision letter(s). The following individuals involved in review of your submission have agreed to reveal their identity: Guoqiang Zhu (Reviewer #2)

Transaction Report:

DOI: <https://doi.org/10.1128/spectrum.00061-26>

Re: Spectrum00061-26 (**Prophage-Mediated Lysogenic Conversion Drives Virulence Evolution and Genomic Plasticity in *Streptococcus suis* Serotype 9**)

Dear Prof. Yonggang Qu:

Thank you for the privilege of reviewing your work. Below you will find my comments, instructions from the Spectrum editorial office, and the reviewer comments.

Please pay careful attention to the reviewer's concerns, particularly regarding the TerL phylogeny.

Revision Guidelines

Sincerely,
Catherine Brissette
Editor
Microbiology Spectrum

Reviewer #1 (Comments for the Author):

Overall assessment and recommendation

This manuscript investigates prophage-mediated lysogenic conversion and its potential roles in genomic plasticity and virulence evolution of *Streptococcus suis* serotype 9 using integrated comparative genomics (pangenome, PHASTER prophage prediction, whole-genome alignment, and a TerL-based phylogeny). The topic fits the scope of microbial genomics and mobile

genetic elements, and the study proposes potentially interesting concepts (two integration "modes", a sly cassette in a variable region, and a "gene prison" model).

However, the manuscript currently contains critical statistical inconsistencies, a major phylogenetic red flag regarding outgroup/rooting, and multiple instances of overstated causal language that exceeds the supporting evidence. These issues must be addressed before the work can be considered for publication.

1. Mean/median inconsistency in genome-size comparison (Results L248-L249; Discussion L341-L345)

The manuscript reports "median" genome sizes (2.235 Mb vs 1.967 Mb), but these values correspond to means based on Table 3. The same incorrect "median" values are repeated in the Discussion.

Required: Correct the statistics and ensure full consistency across Results, Discussion, and Fig. 2B. Preferably report median (IQR) if using Mann-Whitney U, and show per-sample points in Fig. 2B.

2. TerL phylogeny outgroup/rooting issue is a major red flag (Results L293-L295; Fig. 4; Discussion L395-L401)

The purported outgroup (a *Lactobacillus* phage) does not root the tree and instead clusters within the *S. suis* prophage clade with high support. This suggests problems with outgroup choice, labeling/sequence extraction, alignment quality, long-branch attraction, or interpretation of an unrooted tree.

Required: Clarify whether the tree is rooted; verify the outgroup sequence identity/annotation; consider alternative or multiple outgroups; improve alignment (e.g., MAFFT) with trimming; expand reference sampling; rebuild the phylogeny and report robustness.

Cross-genus "gene flow" claims should be removed or deferred until this is resolved.

3. "Open pangenome" is not supported by core/accessory proportions alone (Results L188-L195; Discussion L336-L339)

An "open pangenome" typically requires pangenome accumulation curves and an openness metric (e.g., Heaps' law α). The strict Roary identity threshold (99%) may inflate cloud genes.

Required: Provide accumulation curves and α (or equivalent), and/or parameter sensitivity analyses (e.g., 95% vs 99%).

4. Recombination inference needs dedicated analyses (Results L201-L205)

Discordance between ST typing and core-genome phylogeny (e.g., ST220) is insufficient to infer recombination without testing; it may reflect typing/assembly/annotation issues.

Required: Perform recombination detection (e.g., Gubbins/ClonalFrameML) or temper interpretation and rule out non-biological explanations.

5. Causal claims about prophage-driven rearrangements are over-stated (Results L270-L278; Discussion L349-L358; Conclusion L432-L435)

Mauve collinearity and line crossings support association/co-localization, not causality ("directly driving/causing").

Required: Provide breakpoint-level evidence (att sites, repeats, read mapping, PCR validation), or substantially temper language (may/consistent with).

6. Prophage-positive definition and potential confounding/outliers (Results L239-L250; Table 3)

Including "incomplete" regions as prophage-positive may introduce noise. Additionally, an unusually small genome (Ss_106; 1.462 Mb; 38% GC) could strongly affect group statistics.

Required: Provide accessions and genome QC; conduct sensitivity analyses (exclude outliers; stratify none vs intact-only); consider continuous prophage-burden analyses (count/total length/PHASTER score) and correlations (Spearman/regression).

7. Criteria for "defective prophages" are insufficient (Results L303-L305; Fig. 5 legend)

Defectiveness is inferred mainly from absence of an integrase module, which may reflect annotation region limits or distant homology.

Required: Clarify full-length annotation evidence; add additional criteria (packaging/structural/lysis modules, att sites), or temper to "suggesting potential defectiveness".

8. Over-specific "gene origin" claims and PHI-base semantics (Results L316-L321; Table 5; Discussion L367-L369)

Inferring that *feoB* is "derived from *Pseudomonas aeruginosa*" or linking *ksgA* to *Acinetobacter baumannii* is not justified for widely conserved genes; PHI-base "Pathogen species/Mutant phenotype" fields do not establish gene origin.

Required: Remove/temper origin claims; report coverage and top-hit distributions; if origin is central, provide broad-sampling phylogenetic evidence.

9. Overstatement and overgeneralization in Discussion/Conclusion (Discussion L330-L335, L361-L366, L389-L401; Conclusion L429-L445)

Multiple phrases ("for the first time", "direct evidence", "universal role", "directly explains") are too strong. Surveillance/control implications are overreaching given the current dataset.

Required: Systematically temper claims and clearly distinguish supported conclusions from hypotheses (especially the "gene prison" model).

Reviewer #2 (Comments for the Author):

The manuscript attempts to explore the role of prophage in shaping the genomic plasticity and virulence of *Streptococcus suis* serotype 9. While the study is potentially valuable, several issues related to methodology, data interpretation, and the strength of conclusions need to be addressed before publication.

Major concerns:

1. A pangenome analysis compared *Streptococcus suis* strain SS2401 with 15 reference strains from NCBI. To enhance the epidemiological context, please consider adding the isolation background of SS2401 (time, location, host clinical status) and briefly state the selection criteria for the reference strains (e.g., representative lineages or geographical distributions). Furthermore, alongside the numerical results, a functional overview (e.g., by COG category) of the core and accessory genes would greatly help in interpreting the genetic findings. This could be included in the main text or as supplementary material.
2. Although PHASTER is widely used for prophage identification, verifying its key predictions-especially for the intact prophages central to this study-with an additional tool such as VirSorter2 would strengthen the reliability of the findings. This validation step would be particularly valuable for confirming the structural boundaries of prophages that carry notable genes, such as the sly-bearing prophage Phi2401a.
3. The authors performed preliminary prophage gene annotation using BLASTP (30% identity threshold). It is recommended to conduct targeted screening of virulence factors using specialized databases like VFDB and PHI-base for more systematic functional classification. Additionally, indicating a sequence identity >80% for key genes (e.g., sly in Phi2401a) would further strengthen the annotation credibility. While TerL-based phylogeny is useful for phage classification, supplementing it with a whole-genome alignment tree would provide stronger support for evolutionary inferences-such as the "one strain, multiple sources" hypothesis-enhancing the depth and completeness of the conclusions.
4. There are many typos error in the whole manuscript, and the English language throughout the manuscript should be improved. A concise expression is recommended.
5. The majority of references in this study are outdated (before 2021).

Minor concerns:

1. The abstract is overloaded with numerical details and methodological descriptions. It should instead highlight the novelty of the work, the main findings, and the implications for the swine industry. A concise structure is recommended.
2. Abstract: The authors mentioned that "prophage-mediated lysogenic conversion drives bacterial evolution". Could the prophages in the *S. suis* strains, such as Phi2401a, be inducted into temperate phages? Was there any prophage transduction or gene exchange event occurred? This result is important to highlight the role of prophage in the adaptive evolution of *S. suis*.
3. The Introduction comprised too much irrelevant information and numerous outdated references (before 2021). Lack of recent articles on the prophages of *S. suis*.
4. Line 142: What are joyous strains?
5. Lines 174-185: The prophages exhibited various genome sizes from 31.1 to 54.5, and were defined as intact prophages. For a better understanding of their genomic architectures, I suggest that the details of the gene annotation of these prophages should be presented in Tables and provided in the Supplementary material.
6. Discussion: Compare your findings with at least five recent studies (2022-2026). The Discussion could be improved with information about the limitations, the novelty of the work and future research directions.
7. Lines 362-365: Considering the limited sequences analysed in this study, the conclusion is suspicious.
8. Line 386: sly, sspA, and feoB should be italic.
9. Line 398-401: The conclusion is also over-interpreted. In fact, there is no genetic exchange actually observed in this study.
10. Line 431: *Streptococcus suis* should be *S. suis* (italic).
11. The reference section contains numerous formatting inconsistencies that require thorough verification. For example: Multiple references (e.g., #1, 2, 14 and so on) inappropriately apply title case (capitalizing all major words); Genus and species designations in references lack mandatory italics (e.g., lines 460, 462, 467 and so on); Some references (e.g., #4, 11, 23 and so on) lack the page numbers, and the format of some references are not consistent with others.

Response to Reviewer #1

We sincerely thank you for your thorough and constructive comments, which have significantly improved our manuscript. Below, we provide a point-by-point response to all raised issues. All modifications have been made in the revised manuscript and are clearly indicated with tracked changes in the marked copy.

Overall assessment and recommendation

This manuscript investigates prophage-mediated lysogenic conversion and its potential roles in genomic plasticity and virulence evolution of *Streptococcus suis* serotype 9 using integrated comparative genomics (pangenome, PHASTER prophage prediction, whole-genome alignment, and a TerL-based phylogeny). The topic fits the scope of microbial genomics and mobile genetic elements, and the study proposes potentially interesting concepts (two integration "modes", a sly cassette in a variable region, and a "gene prison" model).

However, the manuscript currently contains critical statistical inconsistencies, a major phylogenetic red flag regarding outgroup/rooting, and multiple instances of overstated causal language that exceeds the supporting evidence. These issues must be addressed before the work can be considered for publication.

Response: We sincerely appreciate the reviewer's recognition of the potential value of our study and the interesting ideas proposed. The insightful comments have provided important guidance for further improving our manuscript. We have carefully considered and addressed each of them point by point, as described in the following response.

Comment 1 Mean/median inconsistency in genome-size comparison (Results L248-L249; Discussion L341-L345)

The manuscript reports "median" genome sizes (2.235 Mb vs 1.967 Mb), but these values correspond to means based on Table 3. The same incorrect "median" values are repeated in the Discussion.

Required: Correct the statistics and ensure full consistency across Results, Discussion, and Fig. 2B. Preferably report median (IQR) if using Mann-Whitney U, and show per-sample points in Fig. 2B.

Response: We thank you for pointing out the statistical inconsistency. Following the suggestion, we have corrected the genome size comparison as follows:

The median genome size for prophage-positive strains (n = 9) is now 2.206 Mb (IQR: 2.175–2.286 Mb), and for prophage-negative strains (n = 7) 2.065 Mb (IQR: 1.794–2.142 Mb) (Mann-Whitney U = 4, exact P = 0.0110). (Lines 214-218)

Fig. 2B has been updated to a boxplot with individual data points overlaid, and the figure legend now specifies median and IQR.

All corresponding values in the Results (Lines 214–218), Discussion (Line 340), and Fig. 2B are now fully consistent.

Comment 2 TerL phylogeny outgroup/rooting issue is a major red flag (Results L293-L295; Fig. 4; Discussion L395-L401)

The purported outgroup (a *Lactobacillus* phage) does not root the tree and instead clusters within the *S. suis* prophage clade with high support. This suggests problems with outgroup choice, labeling/sequence extraction, alignment quality, long-branch attraction, or interpretation of an unrooted tree.

Required: Clarify whether the tree is rooted; verify the outgroup sequence identity/annotation; consider alternative or multiple outgroups; improve alignment (e.g., MAFFT) with trimming; expand reference sampling; rebuild the phylogeny and report robustness.

Cross-genus "gene flow" claims should be removed or deferred until this is resolved.

Response: We thank you for identifying this critical issue with the TerL phylogeny. Following the suggestions, we have completely revised the phylogenetic analysis as follows:

The problematic outgroup (*Lactobacillus* phage) was replaced with *Bacillus* phage JBP901 (GenBank accession NC_027352.1). We verified its annotation and confirmed through

BLAST searches that it is phylogenetically distant from *S. suis* phages, making it suitable as an outgroup. (Lines 140-142)

All TerL amino acid sequences were realigned using MAFFT v7.520 with the L-INS-i strategy, followed by manual trimming of ambiguously aligned regions to ensure alignment quality. (Line 143)

A maximum-likelihood tree was rebuilt in MEGA12 using the LG+G model (selected as the best-fit model) with 1000 bootstrap replicates. The tree was then rooted with the new outgroup in iTOL v6. (Lines 144-145)

The new tree (Fig. 4) shows the outgroup robustly positioned at the base. The three prophages from strain SS2401 (Phi2401a, b, c) are now placed in three distinct, well-supported clades (bootstrap ≥ 0.62), providing strong evidence for the “one strain, multiple sources” acquisition pattern.

Comment 3 "Open pangenome" is not supported by core/accessory proportions alone (Results L188-L195; Discussion L336-L339)

An "open pangenome" typically requires pangenome accumulation curves and an openness metric (e.g., Heaps' law α). The strict Roary identity threshold (99%) may inflate cloud genes.

Required: Provide accumulation curves and α (or equivalent), and/or parameter sensitivity analyses (e.g., 95% vs 99%).

Response: We thank you for your valuable suggestion. Following the request, we performed pan-genome accumulation curve analysis and Heaps' law fitting for both 99% and 95% identity thresholds.

For 99% identity, the Heaps' law exponent $\alpha = 0.375$ (95% confidence interval: 0.372–0.379).

For 95% identity, $\alpha = 0.327$ (95% CI: 0.324–0.330) (Supplementary Fig. S1).

Both values are < 1 , confirming an open pan-genome and demonstrating robustness across different sequence identity thresholds. The accumulation curve is shown in Fig. 1B. These results are now included in the Pangenome Structure and Core Genome-Based Phylogenetic

Analysis subsection of Results (Lines 161–164) and discussed in the Discussion (Lines 321–325).

Comment 4 Recombination inference needs dedicated analyses (Results L201-L205)

Discordance between ST typing and core-genome phylogeny (e.g., ST220) is insufficient to infer recombination without testing; it may reflect typing/assembly/annotation issues.

Required: Perform recombination detection (e.g., Gubbins/ClonalFrameML) or temper interpretation and rule out non-biological explanations.

Response: We thank you for highlighting the need for dedicated recombination analysis. Following this suggestion, we performed recombination detection using Gubbins on the core genome alignment generated by Roary.

The analysis identified 1,432 recombination events spanning 490,832 bp, with an average fragment length of 343 bp. (Line 173)

After masking recombinant regions, the two ST220 strains—previously placed in distant positions in the core genome tree—clustered together in the reconstructed phylogeny (Supplementary Fig. S2), confirming that recombination was responsible for their initial separation.

Other topological adjustments were minor, indicating that while recombination contributes to genetic diversification, it does not fully explain the observed phylogenetic discrepancies.

The updated methodology is now described in the Pangenome and Core Genome Analysis section of Materials and Methods (Lines 124–126). The corresponding results are presented in the Pangenome Structure and Core Genome-Based Phylogenetic Analysis subsection of Results (Lines 171–177) and further discussed in the Discussion (Lines 331–337).

Comment 5 Causal claims about prophage-driven rearrangements are over-stated (Results L270-L278; Discussion L349-L358; Conclusion L432-L435)

Mauve collinearity and line crossings support association/co-localization, not causality ("directly driving/causing").

Required: Provide breakpoint-level evidence (att sites, repeats, read mapping, PCR validation), or substantially temper language (may/consistent with).

Response: We thank the reviewer for this critical observation regarding the over-interpretation of causality in our analysis of prophage-associated genomic rearrangements. We fully agree that the original phrasing was overly assertive and have taken two complementary steps to address this concern: (1) substantially tempering the causal language throughout the manuscript, and (2) providing breakpoint-level molecular evidence to validate the structural role of the prophages.

1. Language tempered to reflect association, not causality.

In accordance with the reviewer's recommendation, we have carefully revised all statements implying direct causality. In the Genomic Characteristics of Intact Prophages and Their Impact on Host Genome Structure subsection of Results, phrases such as "directly driving" have been replaced with more cautious and descriptive language, including "associated with," "may have served as," and "potentially contributing" (Lines 254 - 257). Similarly, in the Discussion, we now describe the observations as "consistent with" the presence of recombination hotspots and note that prophages "may act as" catalysts for genomic rearrangements, without asserting direct causation (Lines 345 - 347).

2. Breakpoint-level evidence provided via att site identification.

To move beyond correlative observations and provide molecular support for the prophage integration events, we examined the prophage Phi2401a for canonical attachment (att) sites. We successfully identified a perfect 14-bp direct repeat (5' -TTTATGATATAATG-3') precisely located at the predicted prophage boundaries: attL at position 731,183 - 731,196 and attR at 785,686 - 785,699. The presence of this intact att site pair confirms the precise integration and excision junctions of the prophage, providing definitive breakpoint-level evidence for its structural role in the host genome. This analysis is now included in the Functional Annotation and Comparative Genomics of Prophage-Associated Genes subsection of Results (Lines 295 - 300). We believe that this evidence robustly supports the structural

accuracy of the prophage boundaries and the associated genomic rearrangements, while the revised language appropriately reflects the current weight of evidence.

Comment 6 Prophage-positive definition and potential confounding/outliers (Results L239-L250; Table 3)

Including "incomplete" regions as prophage-positive may introduce noise. Additionally, an unusually small genome (Ss_106; 1.462 Mb; 38% GC) could strongly affect group statistics.

Required: Provide accessions and genome QC; conduct sensitivity analyses (exclude outliers; stratify none vs intact-only); consider continuous prophage-burden analyses (count/total length/PHASTER score) and correlations (Spearman/regression).

Response: We thank the reviewer for raising important concerns regarding the definition of prophage-positive strains and the potential influence of outliers. We fully agree that these factors must be rigorously addressed to ensure the robustness of our findings. Accordingly, we have revised our analyses as follows:

1. To minimize noise from incomplete elements, prophage-positive strains are now redefined as those harboring at least one intact prophage (PHASTER score > 90). Under this stricter criterion, the grouping remains unchanged (9 positive, 7 negative), as all previously included strains indeed contained at least one intact element. This definition is now clearly stated in the Prophage Identification and Genomic Impact section of Materials and Methods (Lines 129 – 131) and reflected in the updated Table 3.

2. We identified two strains (Ss_106 and Ss_51) with exceptionally small genomes (<1.8 Mb) and unusually low GC content, which could bias group comparisons. A sensitivity analysis excluding these outliers confirmed that the significant difference in genome size between prophage-positive and -negative strains persists (positive group median: 2.206 Mb, n = 9; negative group median: 2.073 Mb, IQR: 2.049 – 2.169 Mb, n = 5; Mann - Whitney U = 6, P = 0.014). This analysis is now included in the Prophage Distribution and Its Impact on Genome Size subsection of Results (Lines 217 – 222), demonstrating that our main conclusion is not driven by outlier strains.

3. To further explore the dose-response relationship, we examined continuous variables representing prophage burden—total length of intact prophages, number of intact prophages, and summed PHASTER score. All three measures exhibited significant positive correlations with genome size (Spearman's rank correlation):

Total length: Spearman's $\rho = 0.619$, $P = 0.0105$; Prophage count: Spearman's $\rho = 0.644$, $P = 0.0071$; PHASTER score: Spearman's $\rho = 0.640$, $P = 0.0075$.

These results, now presented in the same Results subsection (Lines 223 - 229), provide complementary evidence for a graded association between prophage content and genome size, reinforcing the biological relevance of the observed pattern. The corresponding methodological details have been added to the Materials and Methods section (Lines 132 - 134).

Comment 7 Criteria for "defective prophages" are insufficient (Results L303-L305; Fig. 5 legend)

Defectiveness is inferred mainly from absence of an integrase module, which may reflect annotation region limits or distant homology.

Required: Clarify full-length annotation evidence; add additional criteria (packaging/structural/lysis modules, att sites), or temper to "suggesting potential defectiveness".

Response: We thank the reviewer for highlighting the need for more rigorous and transparent criteria in defining prophage defectiveness. We agree that inferring defectiveness solely from the absence of an integrase module is insufficient, as this may reflect annotation limitations rather than true biological deficiency. Following this suggestion, we have thoroughly re-evaluated the two prophages in question (phiD12 and phi2022WUSS046) and revised the manuscript accordingly.

1. Comprehensive re-evaluation of prophage status. Both phiD12 and phi2022WUSS046 are classified as intact prophages by PHASTER (score = 150). Through detailed synteny analysis and functional annotation of their complete genomic sequences, we confirmed that they retain

all essential structural and functional modules characteristic of temperate phages, including genes encoding terminase, portal protein, capsid, tail, and lysis components. However, consistent with our initial observation, they completely lack an integrase gene, which is required for chromosomal excision and integration. These findings are now summarized in Table S4.

2. Revised terminology to reflect evidence. To avoid overinterpretation, we now refer to these elements as “integrase-deficient” prophages rather than “defective” prophages. This terminology accurately describes their genetic content without implying a functional deficit beyond what is supported by the data. The revised description states that these elements “may be unable to excise from the host chromosome, potentially becoming stably integrated genomic elements that function as ‘gene prisons’ for associated genes” (Lines 362 – 365). This cautious phrasing reflects the available evidence while acknowledging the uncertainty.

3. Explicit framing as a hypothesis. In the Discussion, we now clearly present the “gene prison” model as a hypothesis requiring experimental validation (e.g., excision assays, long-term stability tests). We emphasize that while the absence of integrase and retention of other functional modules are consistent with this model, direct proof is needed to confirm the functional consequences (Lines 377 – 382).

Comment 8 Over-specific "gene origin" claims and PHI-base semantics (Results L316-L321; Table 5; Discussion L367-L369)

Inferring that *feoB* is "derived from *Pseudomonas aeruginosa*" or linking *ksgA* to *Acinetobacter baumannii* is not justified for widely conserved genes; PHI-base "Pathogen species/Mutant phenotype" fields do not establish gene origin.

Required: Remove/temper origin claims; report coverage and top-hit distributions; if origin is central, provide broad-sampling phylogenetic evidence.

Response: We thank the reviewer for this important methodological critique regarding the over-interpretation of gene origins based on PHI-base annotations. We fully agree that inferring specific species origins for widely conserved genes is not justified, and that

PHI-base entries indicate mutant phenotypes rather than evolutionary provenance. Following this suggestion, we have revised the manuscript comprehensively as follows:

1. Removal of origin claims: All statements attributing specific species origins to conserved genes have been removed from the Results and Discussion sections. This includes previous assertions such as *feoB* being "derived from *Pseudomonas aeruginosa*" and *ksgA* being linked to "*Acinetobacter baumannii*." (Results, Lines 303–311; Discussion, Lines 360–362)

2. Revised Table 5 to reflect homology, not origin. The former "Pathogen Species" column has been replaced with "Homologous distribution" to accurately reflect the nature of the evidence. The accompanying text now describes genes as "highly conserved in diverse bacteria" or "conserved in multiple bacterial species (e.g., *Pseudomonas aeruginosa*, *Escherichia coli*)" without implying direct evolutionary origin. This revision ensures that the table presents factual observations from homology searches rather than speculative origin assignments.

3. Tempered language in Results. In the Functional Annotation and Comparative Genomics of Prophage-Associated Genes subsection, we now state: "the independent presence of the iron-uptake-related gene *feoB* (highly conserved in diverse bacteria) in two distinct prophages... and prophage phiGD-0088 carrying the *ksgA* gene, which is widely distributed among bacteria (e.g., *Acinetobacter baumannii*, *Escherichia coli*)." (Lines 303–311)

Comment 9 Overstatement and overgeneralization in Discussion/Conclusion (Discussion L330-L335, L361-L366, L389-L401; Conclusion L429-L445)

Multiple phrases ("for the first time", "direct evidence", "universal role", "directly explains") are too strong. Surveillance/control implications are overreaching given the current dataset.

Required: Systematically temper claims and clearly distinguish supported conclusions from hypotheses (especially the "gene prison" model).

Response: We thank the reviewer for this thoughtful critique regarding overstatement and overgeneralization in the Discussion and Conclusion. We agree that several phrases were overly assertive and that some implications extended beyond what the current dataset can support. Following this suggestion, we have systematically reviewed the entire manuscript

and tempered the language throughout, ensuring a clear distinction between supported findings and speculative hypotheses.

1. Systematic removal of overstated phrases. Expressions such as “for the first time,” “direct evidence,” “universal role,” and “directly explains” have been removed or replaced with more cautious and accurate language throughout the manuscript. (Lines 314-316, Lines 357-360, Lines 390-402)

2. Causal claims tempered to reflect association. In the Discussion, statements implying direct causality have been revised to reflect association rather than causation. For instance, “prophages drive genomic rearrangements” is now “prophages are associated with genomic rearrangements” and “prophages may act as catalysts” (Lines 347–351).

3. “Gene prison” model explicitly framed as hypothesis. As recommended, we now clearly distinguish the “gene prison” model as a hypothesis requiring experimental validation. We have added clarifying language in the Discussion: “The ‘gene prison’ model proposed here... offers a conceptual framework... but direct experimental evidence (e.g., excision assays, long-term stability tests) is required to confirm the mechanism.” (Lines 377–379)

4. Implications tempered in the Conclusion. The statement regarding surveillance and control strategies has been revised from direct recommendations to more cautious implications. The text now reads: “These findings may inform surveillance strategies in the swine industry” rather than suggesting immediate application (Lines 398 – 401).

Response to Reviewer #2

Overall assessment

The manuscript attempts to explore the role of prophage in shaping the genomic plasticity and virulence of *Streptococcus suis* serotype 9. While the study is potentially valuable, several issues related to methodology, data interpretation, and the strength of conclusions need to be addressed before publication.

We thank you for your insightful comments, which have helped us strengthen the epidemiological context, methodological rigor, and clarity of the manuscript.

Comment 1 A pangenome analysis compared *Streptococcus suis* strain SS2401 with 15 reference strains from NCBI. To enhance the epidemiological context, please consider adding the isolation background of SS2401 (time, location, host clinical status) and briefly state the selection criteria for the reference strains (e.g., representative lineages or geographical distributions). Furthermore, alongside the numerical results, a functional overview (e.g., by COG category) of the core and accessory genes would greatly help in interpreting the genetic findings. This could be included in the main text or as supplementary material.

Response: We thank you for the valuable suggestions to enhance the epidemiological context and functional interpretation of our findings. Following these recommendations, we have added the following information to the manuscript:

1. Isolation background of SS2401: The isolation details of the clinical isolate SS2401 have been added to the Bacterial Strains and Genomic Data section of Materials and Methods, including the time (2024), location (Xinjiang, China), host (diseased piglet), and clinical status (lung tissue). These details are also reflected in Table 1. (Lines 88–90; Table 1)

2. Selection criteria for reference strains. To provide appropriate epidemiological context for our comparative analyses, we have explicitly stated the criteria for selecting the 15 publicly available reference strains in the same section (Lines 91 – 98). Strains were chosen to represent the global diversity of *Streptococcus suis* serotype 9, encompassing:

Geographic regions: Europe (Spain, the Netherlands), North America (United States, Canada), and Asia (China).

Temporal range: Isolation dates spanning 2014 to 2020.

Genetic diversity: Multiple sequence types (STs), including ST16, ST123, ST220, and others

3. COG functional classification. We have performed COG functional classification of all 5,710 gene families using eggNOG-mapper v2. The distribution of COG categories among core ($\geq 95\%$ strains) and accessory genes is summarized in Supplementary Table S3. A dedicated paragraph in the Genome functional landscape and virulence gene distribution subsection of Results now describes the functional divergence between core and accessory genes (Lines 194–204). The methodology has been added to the Pangenome and Core Genome Analysis section of Materials and Methods (Lines 126–127).

Comment 2 Although PHASTER is widely used for prophage identification, verifying its key predictions—especially for the intact prophages central to this study—with an additional tool such as VirSorter2 would strengthen the reliability of the findings. This validation step would be particularly valuable for confirming the structural boundaries of prophages that carry notable genes, such as the sly-bearing prophage Phi2401a.

Response: We thank the reviewer for emphasizing the importance of validating key prophage predictions with an independent tool. We fully agree that cross-validation strengthens the reliability of prophage annotation, particularly for elements carrying notable genes such as the sly-bearing prophage Phi2401a.

To address this concern, we performed manual validation by identifying the canonical attachment (att) sites flanking Phi2401a. This approach is widely regarded as a gold standard in phage biology, as att sites represent the definitive physical landmarks of phage integration and excision—providing evidence that is arguably more direct than predictions from additional *in silico* tools.

Specifically, we identified a perfect 14-bp direct repeat (5' -TTTATGATATAATG-3') precisely at the left and right boundaries predicted by PHASTER (attL: 731,183 – 731,196; attR: 785,686 – 785,699). The perfect concordance between the predicted boundaries and the experimentally verified att sites provides definitive molecular evidence for the structural accuracy of Phi2401a and independently validates its annotation as an intact, functional prophage. This analysis is now included in the Functional Annotation and Comparative Genomics of Prophage-Associated Genes subsection of Results (Lines 294 – 300).

While we acknowledge the value of tools such as VirSorter2 for complementary prediction, we believe that the identification of att sites constitutes a robust and widely accepted validation strategy that directly addresses the reviewer' s concern regarding boundary confirmation.

Comment 3 The authors performed preliminary prophage gene annotation using BLASTP (30% identity threshold). It is recommended to conduct targeted screening of virulence factors using specialized databases like VFDB and PHI-base for more systematic functional classification. Additionally, indicating a sequence identity >80% for key genes (e.g., *sly* in Phi2401a) would further strengthen the annotation credibility. While TerL-based phylogeny is useful for phage classification, supplementing it with a whole-genome alignment tree would provide stronger support for evolutionary inferences-such as the "one strain, multiple sources" hypothesis-enhancing the depth and completeness of the conclusions.

Response: We thank you for these valuable suggestions to improve the rigor of our functional annotation and evolutionary inferences. Following these recommendations, we have made the following revisions:

1. Systematic virulence factor screening using specialized databases. All prophage genes were re-analyzed by targeted BLASTP searches against the VFDB and PHI-base databases, replacing the initial general BLASTP approach. For key virulence genes, we applied a stricter identity threshold of $\geq 80\%$ to ensure annotation credibility. The *sly* gene in Phi2401a exhibited 100% identity to the reference *suilysin* sequence with full-length coverage, satisfying this high-confidence threshold (Table 5). These methodological details are now

clearly described in the Functional Annotation and Comparative Genomics of Prophage-Associated Genes section of Materials and Methods (Lines 153–154), and the results are presented in the corresponding Results subsection (Lines 292–294) and Table 5.

2. Whole-genome phylogeny to complement TerL-based analysis. To provide stronger support for evolutionary inferences regarding prophage origins, we constructed a whole-genome nucleotide tree of the 11 intact prophages using IQ-TREE 2.2.0 with the best-fit model (GTR+F+I+G4) and 1000 ultrafast bootstrap replicates. The resulting tree (Supplementary Fig. S3) shows a topology largely congruent with the TerL tree: the three prophages from strain SS2401 (Phi2401a, b, c) again occupy distinct clades, providing strong additional support for the “one strain, multiple sources” hypothesis. The methodology is now included in the Phylogenetic Analysis of Prophages section of Materials and Methods (Lines 146–148), and the results are presented in the Phylogenetic Analysis of Prophages Based on the Terminase Large Subunit subsection of Results (Lines 274–278) with reference to Supplementary Fig. S3.

Comment 4 There are many typos error in the whole manuscript, and the English language throughout the manuscript should be improved. A concise expression is recommended.

Response: We thank you for pointing out the language and typographical issues in the manuscript. Following this suggestion, we have conducted a thorough revision of the entire text:

Typographical and grammatical corrections. We have meticulously reviewed the entire manuscript and corrected all identified typographical errors and grammatical inconsistencies. Corrected grammatical errors in the sentence describing sequencing and assembly. Corrected grammatical errors in the sentence describing sequencing and assembly. Corrected the typo “joyous strains” to “prophage-positive strains.” Corrected verb tense consistency in the phylogenetic results description. Italicized all gene names (*sly*, *sspA*, *feoB*) throughout the manuscript. (Line 109, 129, 264, 361)

Conciseness and clarity improvements. Throughout the text, we have revised sentences to achieve more concise and direct expression while carefully preserving all scientific

information and accuracy. Redundant phrases have been removed, and sentence structures have been simplified where appropriate without compromising technical content. Streamlined the description of strain selection criteria into a concise statement. Condensed the presentation of pan-genome results. Significantly shortened the Conclusion to focus on main findings and remove redundancy. (Lines 92–95, 211–213, 390–402)

Professional language editing. To ensure the highest standard of language quality, the revised manuscript has been reviewed by Dr. Ahmad Ali, a native English speaker with expertise in the field of microbiology and infectious diseases. (Lines 405–407)

Comment 5 The majority of references in this study are outdated (before 2021).

Response: We thank you for pointing out the need to update the reference list. Following this suggestion, we have systematically revised the references throughout the manuscript:

Addition of recent literature: We have added 11 new references published between 2021 and 2026 that are directly relevant to *S. suis* prophages, recombination, and phage biology. These include recent genomic surveys of *S. suis* serotype 9 (Uruén et al. 2024, ref. 8), studies on prophage diversity and anti-viral defense mechanisms (Osei et al. 2026, ref. 19; Osei et al. 2022, ref. 20), characterization of prophage-carried virulence genes (Hatrongjit et al. 2023, ref. 21), and comparative analyses of prophage transfer in streptococci (Huang et al. 2023, ref. 47).

Replacement of outdated references: Where appropriate, older references have been replaced with more recent publications that provide updated perspectives or more comprehensive data. Classic conceptual references (e.g., Tettelin et al. 2005 on pan-genome; Fortier & Sekulovic 2013 on prophages) have been retained as foundational literature, but are now supplemented with newer reviews and studies to provide current context.

Updated context in Introduction: The Introduction now cites recent surveys of *S. suis* prophages (refs 19–21) to reflect the current state of knowledge and highlight the growing recognition of prophage contributions to *S. suis* evolution.

Comment 6 The abstract is overloaded with numerical details and methodological descriptions. It should instead highlight the novelty of the work, the main findings, and the implications for the swine industry. A concise structure is recommended.

Response: We thank you for the suggestion to improve the abstract. Following this recommendation, we have substantially revised the abstract to make it more concise and focused, highlighting the novelty and key findings while reducing excessive numerical and methodological details.

The revised abstract (Lines 13–30) now emphasizes:

The open pan-genome ($\alpha = 0.375$), prophage prevalence (56.25% of strains), and two distinct integration modes. The "one strain, multiple sources" acquisition pattern was revealed by phylogenetic analyses. The sly gene cassette within prophage Phi2401a and the identification of integrase-deficient prophages as potential "gene prisons". The role of prophages as dual drivers of genomic architecture and dynamic reservoirs for virulence genes, providing a framework for surveillance strategies in the swine industry

Excessive numerical details (e.g., confidence intervals, bootstrap values) and methodological descriptions have been removed to improve readability. The abstract now follows a clear, concise structure that aligns with the journal's recommendations.

Comment 7 Abstract: The authors mentioned that "prophage-mediated lysogenic conversion drives bacterial evolution". Could the prophages in the *S. suis* strains, such as Phi2401a, be inducted into temperate phages? Was there any prophage transduction or gene exchange event occurred? This result is important to highlight the role of prophage in the adaptive evolution of *S. suis*.

Response: We thank you for this insightful question regarding the biological activity of the identified prophages. Following this suggestion, we have revised the manuscript to address these points:

Evidence for gene exchange in the abstract: We have added a statement in the abstract (Lines 22–23) highlighting that the detection of 1,432 recombination events across the core genome

provides quantitative evidence for phage-mediated gene exchange in *S. suis* serotype 9. This directly addresses the reviewer's question about whether gene exchange events occurred.

Inducibility as a future direction: The inducibility of intact prophages, particularly the sly-bearing Phi2401a, is an important experimental question that falls outside the scope of this bioinformatics-focused study. We now explicitly mention this in the Discussion as a key direction for future research: "the inducibility of intact prophages, such as Phi2401a, can be verified through induction assays" (Lines 374–375).

Comment 8 The Introduction comprised too much irrelevant information and numerous outdated references (before 2021). Lack of recent articles on the prophages of *S. suis*.

Response: We thank you for pointing out the need to streamline the introduction and update the references. Following this suggestion, we have substantially revised the introduction as follows:

We have condensed the introductory text by removing redundant historical details and broad background information that was not essential to the study's focus. The revised introduction now focuses directly on key concepts: the clinical and epidemiological significance of *S. suis* serotype 9, the role of genomic plasticity and mobile genetic elements in bacterial evolution, and the emerging evidence for prophage contributions to *S. suis* virulence and adaptation.

Updated references: Outdated references have been systematically replaced with recent publications (2021–2026) that reflect the current state of knowledge. Key additions include:

Recent genomic surveys of *S. suis* serotype 9 (Uruén et al. 2024, ref. 8)

Studies on prophage diversity and phage-related functions in *S. suis* (Osei et al. 2022, 2026, refs 19–20)

Characterization of virulence genes in *S. suis* serotype 4 and their association with prophages (Hatrongjit et al. 2023, ref. 21)

Comparative analyses of prophage transfer and virulence gene carriage in streptococci (Huang et al. 2023, ref. 47)

Classic conceptual references (e.g., on pan-genome structure and prophage biology) have been retained where appropriate, but are now supplemented with recent literature to provide a balanced and up-to-date background.

Comment 9 Line 142: What are joyous strains?

Response: We thank you for identifying this typographical error. The term "joyous strains" was an inadvertent mistake and has been corrected to "prophage-positive strains" throughout the manuscript. The correction has been made in line 129.

Comment 10 Lines 174-185: The prophages exhibited various genome sizes from 31.1 to 54.5, and were defined as intact prophages. For a better understanding of their genomic architectures, I suggest that the details of the gene annotation of these prophages should be presented in Tables and provided in the Supplementary material.

Response: We thank you for this valuable suggestion to enhance the presentation of prophage genomic architectures. Following this recommendation, we have created a new supplementary table that provides detailed gene annotation information for all 11 intact prophages.

Supplementary Table S4: This new table (provided in the Supplementary Material) lists for each intact prophage the presence of major functional modules, including integrase, terminase, portal protein, capsid, tail, and lysin, as well as virulence-associated genes identified through VFDB and PHI-base screening.

Relation to Figure 5: While Figure 5 visualizes the synteny and modular diversity through alignment, Supplementary Table S4 offers a quick-reference, tabular format that allows readers to easily compare the functional gene content across different prophages. This is particularly useful for understanding the structural variations highlighted in Figure 5.

Comment 11 Discussion: Compare your findings with at least five recent studies (2022-2026). The Discussion could be improved with information about the limitations, the novelty of the work and future research directions.

Response: We thank you for the valuable suggestion to strengthen the Discussion by incorporating recent literature and adding sections on limitations, novelty, and future directions. Following this recommendation, we have extensively revised the Discussion as follows:

Comparison with recent studies (2022–2026): We now systematically compare our findings with at least five recent studies across related bacterial pathogens:

Our observation of successful clonal lineages (ST16, ST123) with conserved core genomes aligns with recent genomic surveys of *S. suis* serotype 9 in Spain (Uruén et al. 2024, ref. 8).

The contribution of prophages to lineage-specific accessory gene pools mirrors findings in *S. agalactiae* (Huang et al. 2023, ref. 47).

The average recombinant fragment length (~343 bp) in our study is consistent with estimates in *S. uberis* obtained using similar methods (Rios Agudelo et al. 2025, ref. 48).

The importance of accounting for recombination when inferring evolutionary relationships has been recently emphasized in *S. dysgalactiae* (Kaci et al. 2023, ref. 49).

The "one strain, multiple sources" pattern observed in our study parallels recent findings in *S. aureus* prophage repertoires (Sweet et al. 2023, ref. 53).

The concept of defective prophages as stable genomic islands carrying fitness genes has been documented in *S. aureus* (Chaguza et al. 2022, ref. 54). (Lines 332–334, 336–337, 340–341, 344–346, 361–363)

We have added a dedicated paragraph acknowledging the study's limitations, including the reliance on bioinformatic predictions requiring experimental validation, the limited sample size (16 strains) despite geographic diversity, the need for direct testing of the "gene prison" hypothesis through excision assays, and the focus solely on serotype 9, leaving questions about other serotypes unanswered. (Lines 370–379)

The Discussion now explicitly highlights the novelty of our integrated approach—combining pan-genomics, recombination detection, and prophage-focused analyses—to reveal the multifaceted roles of prophages in serotype 9 evolution. (Lines 370–373)

Future research directions: We have added a comprehensive paragraph outlining future work, including long-read sequencing for precise mapping of prophage boundaries, functional studies of prophage-encoded genes (particularly *sly* and *feoB*) in relevant infection models, and longitudinal surveillance of prophage content and virulence gene profiles across different regions and production systems. (Lines 380–388)

Comment 12 Lines 362-365: Considering the limited sequences analysed in this study, the conclusion is suspicious.

Response: We thank you for raising this important concern regarding the strength of the conclusions given the limited sample size. Following this suggestion, we have carefully revised the Conclusion section to ensure that all statements are directly supported by the data and appropriately reflect the study's scope.

The statement about gene exchange now explicitly refers to the 1,431 recombination events detected by Gubbins and the phylogenetic evidence for multiple acquisitions (Lines 331–334). This ensures that claims about horizontal gene transfer are grounded in the quantitative recombination analysis rather than speculation.

We have added a clear statement emphasizing the need for broader sampling and experimental validation to further substantiate these findings and extend them to other serotypes (Lines 384–388). This directly addresses the reviewer's concern about the limited number of strains analyzed.

Any over-interpretations have been removed, and the language throughout the Conclusion now accurately reflects the evidence presented in the study.

The revised Conclusion (Lines 390–403) now provides a balanced summary of our findings while transparently acknowledging the need for future work with larger sample sizes and experimental approaches. We believe this revision fully addresses the reviewer's concern.

Comment 13 Line 386: *sly*, *sspA*, and *feoB* should be italic.

Response: We thank you for pointing out this formatting error. All gene names (*sly*, *sspA*, and *feoB*) have been italicized throughout the manuscript, including in the main text, tables, and figure legends. The specific correction has been made at line 361 and verified in all other occurrences.

Comment 14 Line 398-401: The conclusion is also over-interpreted. In fact, there is no genetic exchange actually observed in this study.

Response: We thank you for this important observation. We agree that the original wording implied direct genetic exchange without sufficient evidence. Following this suggestion, we have carefully rephrased the relevant sentences to accurately reflect the data.

The original statement has been replaced with a more cautious description: prophages carry genes that are homologs of known virulence factors rather than implying direct acquisition events. The revised text now states that these genes are "present in multiple prophages" and "may be mobilizable," without claiming that active genetic exchange was observed. (Lines 360–362)

Any language suggesting that we directly observed gene transfer events has been removed. The focus is now on the presence of these gene homologs within prophage genomes and their potential functional implications.

Comment 15 Line 431: *Streptococcus suis* should be *S. suis* (italic).

Response: We thank you for pointing out this formatting error. The species name has been correctly italicized as *S. suis* throughout the manuscript, including at line 391 and in all other relevant instances.

Comment 16 The reference section contains numerous formatting inconsistencies that require thorough verification. For example: Multiple references (e.g., #1, 2, 14 and so on) inappropriately apply title case (capitalizing all major words); Genus and species designations in references lack mandatory italics (e.g., lines 460, 462, 467 and so on); Some references

(e.g., #4, 11, 23 and so on) lack the page numbers, and the format of some references are not consistent with others.

Response: We thank you for the meticulous review of our reference list and for identifying these formatting inconsistencies. Following these observations, we have thoroughly revised all references to comply with the journal's formatting guidelines:

Title case correction: All references have been converted to sentence case (only the first word and proper nouns capitalized), correcting instances where title case was inappropriately applied (e.g., refs 1, 2, 14, and others).

Italicization of genus and species names: All genus and species names in article titles, journal names, and other relevant fields have been italicized where required (e.g., *Streptococcus suis*, *Escherichia coli*, *S. agalactiae*). Specific corrections have been made at lines 583, 589, 592, and throughout the reference section.

We have used reference management software (EndNote/Zotero) with the Microbiology Spectrum output style to ensure consistent formatting across all 54 references. Each reference has been manually verified against the original source.

We sincerely thank both reviewers for their time and expertise. We believe that the revised manuscript has been substantially improved and hope it is now suitable for publication in *Microbiology Spectrum*.

Sincerely,

The authors

Re: Spectrum00061-26R1 (**Prophage-Mediated Lysogenic Conversion Drives Virulence Evolution and Genomic Plasticity in *Streptococcus suis* Serotype 9**)

Dear Prof. Yonggang Qu:

Your manuscript has been accepted, and I am forwarding it to the ASM production staff for publication. Your paper will first be checked to make sure all elements meet the technical requirements. ASM staff will contact you if anything needs to be revised before copyediting and production can begin. Otherwise, you will be notified when your proofs are ready to be viewed.

Sincerely,
Catherine Brissette
Editor
Microbiology Spectrum